# Can Enterocin M in Combination with Sage Extract Have Beneficial Effect on Microbiota, Blood Biochemistry, Phagocytic Activity and Jejunal Morphometry in Broiler Rabbits?

**DOI:** 10.3390/ani10010115

**Published:** 2020-01-10

**Authors:** Monika Pogány Simonová, Ľubica Chrastinová, Anna Kandričáková, Soňa Gancarčíková, Eva Bino, Iveta Plachá, Jana Ščerbová, Viola Strompfová, Rudolf Žitňan, Andrea Lauková

**Affiliations:** 1Institute of Animal Physiology, Centre of Biosciences of Slovak Academy of Sciences, Šoltésovej 4-6, 04001 Košice, Slovakia; kandricakova@saske.sk (A.K.); glatzova@saske.sk (E.B.); placha@saske.sk (I.P.); imrichova@saske.sk (J.Š.); strompfv@saske.sk (V.S.); laukova@saske.sk (A.L.); 2National Agricultural and Food Centre, Hlohovecká 2, 951 41 Nitra-Lužianky, Slovakia; chrastinova@vuzv.sk (Ľ.C.); zitnan@vuzv.sk (R.Ž.); 3Laboratory of Gnotobiology, Department of Microbiology and Immunology, University of Veterinary Medicine and Pharmacy, Komenského 73, 041 81 Košice, Slovakia; sona.gancarcikova@uvlf.sk

**Keywords:** enterocin, microbiota, morphometry, phagocytic activity, rabbit, sage

## Abstract

**Simple Summary:**

The weaning period is one of the most critical phases in a rabbit’s life, when rabbits are most sensitive to various spoilage agents, often with fatal outcome. Preventing these post-weaning alimentary disturbances and improving gut health through the use of natural substances has become an area of research activity in rabbit nutrition. The possibilities of “in vivo” administration of bacteriocins in rabbits are often limited. Our study focused therefore on improving rabbits’ health using enterocin M and sage extract, alone and also in combination, mainly to determine their effect on the microbiota, phagocytic activity, serum biochemistry, and morphometry of rabbits. The antibacterial effect of both additives was observed in the gastrointestinal tract of rabbits. Several biochemical parameters (increased in total proteins (TPs) using enterocin (Ent) M, reduction of alanine aminotransferase (ALT) using sage and its hypocholesteremic effect alone and in combination with EntM, and improvement of mineral levels) were optimized. All additives, but mostly EntM, showed a tendency to improve the jejunal morphological parameters.

**Abstract:**

The present study investigated the effects of enterocin (Ent) M and sage extract alone and also in combination on the gut microflora, phagocytic activity, blood biochemistry, and morphometry of rabbits. Sixty-four rabbits (aged five weeks, M91 meatline, both sexes) were divided into three experimental groups: E (EntM; 50 µL/animal/day), S (sage; 10 µL/animal/day), and E + S (EntM + sage) groups and control group (C). The additives were administered in drinking water for a period of 21 days. Dietary supplementation of EntM and sage significantly reduced the coliforms (E: *p* < 0.001; S: *p* < 0.001; E + S: *p* < 0.001) in feces, while simultaneous addition of EntM and sage decreased enterococci (E + S: *p* < 0.0001), lactic acid bacteria (E + S: *p* < 0.01), and coagulase-positive staphylococci (E + S: *p* < 0.0001) in the appendix. Sage addition reduced HDL (S: *p* < 0.001) and LDL cholesterol (S: *p* < 0.001; E + S: *p* < 0.001), LDL/HDL ratio (S: *p* < 0.001; E + S: *p* < 0.01), and increased urea (S: *p* < 0.01; E + S: *p* < 0.001) and creatinine (S: *p* < 0.001; E + S: *p* < 0.001) in serum. EntM and sage application, alone or in combination, improve the jejunal morphometry (*p* < 0.0001) in rabbits.

## 1. Introduction

Rabbit breeding is an important enterprise in many countries, because these animals provide high quality meat for human consumption. Rearing rabbits can cause many environmental, dietary, and physiological stresses, mainly during the weaning period. These 10–15 days are the most critical period for rabbits as it is when they are most sensitive to various spoilage agents, often with fatal outcome. For these reasons, preventing post-weaning alimentary disturbances, maintaining the rabbits’ health, and stabilizing the economy of breeding is still a hot topic. To achieve better production in rabbit husbandries, one possible way is the use of natural compounds, such as probiotics, prebiotics, synbiotics, bacteriocins, organic and fatty acids, and plant extracts, which can improve the health status of animals [1,2,3,4]. There are many reviews presenting their beneficial effects, mostly of probiotic applications to rabbits, with emphasis on reduction of pathogens and parasites in the gastrointestinal tract (GIT), stimulation of immunity, better growth performance, metabolism and nutrient utilization, and improved meat quality [5,6,7]. Application of natural antimicrobials, including probiotics, bacteriocins, and herbal extracts, can modulate and balance the intestinal microbiota instead of beneficial lactic acid bacteria, and enhance the immune response. The mechanisms of action of probiotics are well known: Reduction of toxin production, stimulation of enzyme production by the host, competition for adhesion to epithelial cells, stimulation of the immune system of the host (mainly due to stimulation of gut-associated lymphoid tissue (GALT) and mucosa-associated lymphoid tissue (MALT)), decreasing intestinal pH, and production of vitamins (B, K) and antimicrobial substances (lactic and acetic acids, hydrogen peroxide, carbon dioxide, diacetyl, and bacteriocins [8,9]). Bacteriocins, defined as ribosomally synthesized antimicrobial peptides, are substances of proteinaceous character with a broad antimicrobial spectrum, produced mostly by lactic acid bacteria such as lactobacilli, lactococci, pediococci, and enterococci [10]. They have traditionally been used as food preservatives, either added or produced by starter cultures during fermentation. Moreover, the most often used bateriocins, nisin and pediocin, have been used in other ecosystems: Feed (silage), animals, aquacultures, and also humans [11,12,13]. Enterocins represent an individual group of bacteriocins which are produced by different species of enteroccocci, but mainly by *Enterococcus faecium*, *E. faecalis*, *E. hirae*, *E. durans*, and *E. mundtii* [10,14]. The antimicrobial effects of bacteriocins and sage extract are well known, including mostly Gram-positive but also Gram-negative bacteria. Although there are many studies about enterocin application in the food industry and humans, possibilities of their “in vivo” administration in animals, including rabbits, are often limited [15]. A very similar situation is present in the case of plant extract inclusion in rabbit husbandries; studies of their effects on rabbits are more or less preliminary [16,17,18,19]. Sage (*Salvia* spp.) is a common aromatic and medicinal plant encompassing about 900 species belonging to the mint family Lamiaceae (*Labiatae*), and is attributed with antioxidant, anti-inflammatory, antimicrobial, hypocholesteraemic, hypoglycaemic, and anti-mutagenic bioactivities. Many *Salvia* spp. are used in herbal tea, food flavoring, cosmetics and perfumeries, and in the pharmaceutical industry. There is also growing interest in sage plants, seeds, and extracts used in animal feeding due to their oil content, which is a rich source of polyunsaturated fatty acids (PUFA) (linoleic and α-linolenic acid). Dietary administration of sage, its extracts and/or by-products could also improve the PUFA content of animal products (eggs, meat [1]), and so there is growing effort and interest to continuously complement these studies in rabbit farms.

The objectives of this study were to determine the effects of enterocin (Ent) M and sage extract, alone and also in combination, on selected parameters: Microbiota, phagocytic activity, serum biochemistry, and morphometry of rabbits.

## 2. Materials and Methods

### 2.1. Animals and Housing

Sixty-four rabbits (meat line M91, aged 35 days after weaning, both sexes both sexes: 60 male and four female rabbits) were randomly divided into three experimental groups (E–EntM, S–sage extract, and E + S–EntM in combination with sage extract) and one control group (C) with 16 animals in each group. The average live weights of the rabbits at the start of the experiment were 940.6 ± 180.8 g in E; in S it was 993.1 ± 300.0 g; in E + S it was 1002.3 ± 162.3 g; and in C it was 1042.5 ± 315.7 g. The rabbits were kept in standard cages (0.61 m by 0.34 cm by 0.33 m, type D-KV-72 supplied by the Kovobel company, Domažlice, Czech Republic), two animals per cage. The cages allowed feces separation. A cycle of 16 h light and 8 h dark was used throughout the experiment. Temperature 20 ± 4 °C and humidity 70% ± 5% were maintained throughout the experiment by heating and ventilation systems, and data were recorded continuously with a digital thermograph positioned at the same level as the cages. The experiment was performed in co-operation with our colleagues in Nitra (National Agricultural and Food Centre—NAFC). All care and experimental procedures involving animals followed the guidelines stated in the Guide for the Care and Use of Laboratory Animals approved by the State Slovak Veterinary and Food Administration and the Ethics Committees of both institutions (permission code: SK CH 17016 and SK U 18016).

### 2.2. Experimental Design Slaughtering and Sampling

The animals were fed an untreated diet (pellets of 3.5 mm in average size), commonly used in the nutrition of growing rabbits. The ingredients and chemical composition of this diet is presented in Table 1. The ME content was calculated using the equation of Wiseman et al. [20]. Chemical analyses were conducted according to the Association of Official Agricultural Chemists (AOAC) [21] with the considerations given by Gidenne et al. [22] for dry matter (DM), crude protein (CP), crude fiber (CF), crude fat, nitrogen free extract, ash, and organic matter. Neutral detergent fiber (NDF) and acid detergent fiber (ADF) were analysed sequentially (Van Soest et al. [23]) with a thermo stable amylase pre-treatment and starch according to the alpha-amyloglucosidase method.

The rabbits in group E were administered EntM (prepared according to Mareková et al. [24]; dose 50 µL/animal/day, with activity 12,800 AU/mL, in concentration 0.8 g/L, from day zero to day 21) in their drinking water. Activity of EntM was tested with the agar spot test according to De Vuyst et al. [25] against the principal indicator strain *E. avium* EA5 (isolated from piglet feces in our laboratory). Rabbits in group S received sage plant extract (*Salvia officinalis* extract containing 24% thujone, 18% borneol, and 15% cineole; Calendula company, Novaá Ľubovňa, Slovakia) in their drinking water at a dose of 10 μL/animal/day for 21 days. The animals in group E + S were administered (from day zero to day 21) a combination of EntM (50 μL/animal/day, 12,800 AU/mL, in concentration 0.8 g/L) and sage extract (10 μL/animal/day, in concentration 0.15 g/L). The doses of additives and their manner of application were decided on the basis of our previous in vitro studies testing the inhibitory activity of EntM and sage extract against target bacteria [19] and an experiment with rabbit-derived bacteriocin-producing strain *E. faecium* EF2019 (CCM7420 [4]). Based on our previous experiments, these additives can be dissolved in distilled water and/or phosphate buffer [19,24] and also, we had information about the volume of water drunk by rabbits; the additives were applied firstly to 100 mL of drinking water in all cages, and after consuming this volume the rabbits had access to water ad libitum. Control rabbits (group C) had the same conditions, but without additives being applied to their drinking water, and they were fed a commercial diet. Drinking water was provided through nipple drinkers. The experiment lasted for 35 days.

Body weight and feed consumption were measured every week during the experiment; average daily weight gain and feed conversion were calculated mathematically. Mortality and morbidity were also recorded in groups daily. Freshly-voided feces were collected using nets mounted under the cages (two to three nets/group). At the start of the experiment (at day zero, without additives inclusion), we decided to collect 10 mixed samples from all experimental and control groups—initial microbial background). Because there were two animals housed in each cage (eight cages), at days 21 (three weeks of additives application) and 35 (end of the experiment, two weeks of additives cessation) we collected one sample from under each cage, which were eight samples per net, i.e., per group. Blood was sampled from eight rabbits, individually from each of them, at days 21 and 35 (*n* = 8/group). At days 21 and 35, rabbits were randomly selected for slaughter (*n* = 8), appendix and jejunum were sampled to test microbial profile in appendix and morphometry testing in jejunum. Rabbits were stunned with electronarcosis (90 V for 5 s), immediately hung by the hind legs on the processing line and quickly bled by cutting the jugular veins and the carotid arteries.

### 2.3. Microbial Isolation and Analysis

To test microbiota, the samples of feces and appendix content (1 g) were treated using the standard microbiological diluton method (International Organization for Standardization (ISO)). The appropriate dilutions in Ringer solution (pH 7.0; Oxoid Ltd., Basingstoke, Hampshire, England) were plated onto following media: M-Enterococcus agar (NF-V04503, Difco Laboratories, Detroit, MI, USA) for enterococci, DeMann-Rogosa-Sharpe agar (ISO 15214, Merck, Germany) for lactic acid bacteria (LAB), mannitol salt agar for coagulase-negative staphylococci (CoNS, ISO 6888), Baird-Parker agar enriched with egg yolk tellurite supplement (ISO 21527-1, Difco) for coagulase-positive staphylococci and S. aureus (CoPS), Clostridium difficile agar with the supplement SR0096E 7% (*v*/*v*) defibrinated horse blood (SR0050, ISO 15883, Oxoid) for *Clostridium* species (anaerobic cultivation), MacConkey agar (ISO 7402, Oxoid) for coliforms, and CLED agar (Conda, Spain) for enterobacteria. Pseudomonads were isolated on Pseudomonas agar (Biomark, India). Cultivation was performed at 30 °C and/or 37 °C for 24–48 h depending on the bacterial genera. The bacterial counts were expressed in log 10 of colony forming units per gram (log 10 CFU/g ± SD). Randomly picked up representants of selected bacterial groups were confirmed by MALDI-TOF identfication system (Bruker Daltonics).

### 2.4. Blood Analysis Phagocytic Assay and Morphometry Testing

Blood (*n* = 8) was sampled from the marginal ear vein (*Vena auricularis*) into dry non-heparinized Eppendorf tubes at days 21 and 35 for biochemical analyses. Blood serum was received by centrifugation at 3000× *g* for 10 min and stored frozen (at −18 °C) in plastic vials until analysis. To test the biochemical parameters—total proteins (TP; g/L), albumins (g/L), creatinine (µmol/L), urea (mmol/L), glucose (mmol/L), triglycerides (mmol/L), total cholesterol (mmol/L), HDL cholesterol (mmol/L), LDL cholesterol (mmol/L), LDL/HDL ratio, alanine aminotransferase (ALT; µkat/L), aspartate aminotransferase (AST; µkat/L), gamma glutamyl transferase (GMT; µkat/L), alkaline phosphatase (ALP; µkat/L), α-amylase (µkat/L), pancreatic amylase (µkat/L), calcium (mmol/L), phosphorus (mmol/L), magnesium (mmol/L), and chlorids (mmol/L), a commercial kit Dialab (Cezh Republic) and an automated biochemical analyzer ELLIPSE (AMS, Italy) based on turbidimetric clinical assay were used, according to the manufacturer´s instructions and following the methods of serum parameters determinations (www.amsalliance.com/it, [26,27,28,29,30,31,32,33,34,35,36,37,38]). The activity of blood glutathione-peroxidase (GPx; µkat/L) was determined by the colorimetric method (Spectrophotometer UV-2550 Shimadzu, Kyoto, Japan) using commercial kit Randox RS 504 (Randox Laboratories Ltd., Crumlin, UK).

For phagocytic activity (PA), it was sampled into Eppendorf tubes containing microsferic hydrophilic (MSH) particles and heparin. Phagocytic activity was measured by a direct microscopic counting procedure, using MSH particles according the modified test described by Vetvička et al. [39]: 50 µL of MSH particle suspension (ARTIM, Prague, Czech Republic) was mixed with 100 µL of blood in an Eppendorf-type test tube and incubated at 37 °C for 1 h. Blood smears were then prepared and stained in accordance with May-Grünwald and Giemsa-Romanowski. PA was calculated as the number of white cells containing at least three engulfed particles per 100 white cells (monocytes/granulocytes). The index of phagocytic activity (IPA) was calculated as the number of engulfed particles per total number of phagocytes observed.

To test morphometry (villus cut surface, villus circumference, villus height (VH), crypt depth (CD), and villus height:crypt depth ratio—VH:CD), intestinal tissue (1 cm^2^) of the proximal jejeunum was sampled and treated as previously described by Žitňan et al. [40]. Briefly, intestinal tissue (1 cm^2^) from proximal jejunum were fixed in 4% neutral formaldehyde solution. After being rinsed in water, the samples were dehydrated in a graded series of ethanol (30%, 50%, 70%, 90%, and absolute ethanol), cleared in benzene, saturated with and embedded in paraffin. Sections of 5 µm thickness (10 slices of each sample) were stained with haematoxylin/eosin and observed under a light microscope. The height, circumference, and cut surface area of 30 villi and depth of 30 crypts were determined by the computer-operated *Image C* picture analysis system (Imtronic GmbH, Berlin, Germany) and the interactive measurement (IMES) analysis programme, by using a color video camera (SONY 3 CCD, Sony Electronics Ltd., Tokyo, Japan) and a light microscope (Axiolab, Carl Zeiss AG, Jena, Germany).

### 2.5. Statistical Analysis

Treatment effects on tested parameters were analysed using two-way analysis of variance (ANOVA), followed by a Bonferroni post-hoc test for pair-wise comparisons, where appropariate. The statistical model included the time and treatment effects and their interactions. All statistical analyses were performed using GraphPad Prism statistical software (GraphPad Prism version 6.0, GraphPad Software, San Diego, CA, USA). Differences between the mean values of the different dietary treatments were considered statistically significant at *p* < 0.05. Data are expressed as means and pooled standard deviations of the mean (SD).

## 3. Results

### 3.1. Bacterial Populations

All animals were found to be in good health condition throughout the trial. In feces, the bacterial counts (except *Clostridium*-like sp.) were influenced by time. The effect of treatment was observed only in the coliform and pseudomonads; results of treatment with x time interaction were also found to be significant in these groups of bacteria (Table 2). Significant reduction in CoNS (E: *p* < 0.05), CoPS (S: *p* < 0.01; E + S: *p* < 0.01), and coliforms (day 21; E: *p* < 0.001; S: *p* < 0.001; E + S: *p* < 0.001) was noted in fecal samples compared to C during EntM and sage application on day 21. After additives cessation (day 35), lower counts of enterobacteria (E + S: *p* < 0.05) were detected compared with group C. In the appendix the bacterial counts of enterococci, CoPS, and coliforms were affected by the treatment, with significantly reduced enterococci (S: *p* < 0.001; E + S: *p* < 0.001) and CoPS (S: *p* < 0.001; E + S: *p* < 0.001), while time influenced only LAB and clostridia (Table 3). Interaction of treatment x time was also identified for most of the tested bacterial groups.

### 3.2. Serum Chemistry Phagocytic Activity and Jejunal Porphometry

All the tested parameters were influenced by the treatment, time, and interactions of treatment and time, except GMT and lipase (Table 4 and Table 5). The most striking differences in lipid metabolism were observed during the administration of sage extract alone, but also in combination with EntM; in this regard, sage extract alone or together with EntM reduced the level of HDL cholesterol (S: *p* < 0.001; E + S: *p* <0.05), LDL cholesterol (S: *p* < 0.001; E + S: *p* < 0.001), and LDL/HDL ratio (S: *p* < 0.001; E + S: *p* < 0.01) in comparison with control data. Moreover, lower LDL cholesterol level and LDL/HDL ratio were still evident in groups S (*p* < 0.01; *p* < 0.001) and E + S (*p* > 0.05; *p* < 0.001) even two weeks after additives cessation (day 35). In contrast to EntM, sage extract significantly increased urea (S: *p* < 0.01; E + S: *p* < 0.001) and creatinine (S: *p* < 0.001; E + S: *p* < 0.001) levels in the blood, which demonstrates its negative impact on the blood nitrogen profile due to worsening of glomerular filtration in the kidneys. The effect of sage on albumin concentration was interesting: While during its application the albumin levels in the blood remained high, after its cessation these concentrations were significantly reduced (S: *p* < 0.05; E + S: *p* < 0.001). The lowest glucose level was measured in E + S (*p* < 0.001) compared to other goups, during the whole experiment. Concerning the hepatic and enzymatic profile, the most striking changes were recorded in the group E + S; the levels of AST (*p* < 0.001) were found to be the highest, while ALP (*p* < 0.001), α-amylase (*p* < 0.001), and pancreatic amylase (*p* < 0.001) concentrations decreased to the lowest value. Among the tested minerals, the most significant differences were observed in the case of combined administration of sage and enterocin (E + S); the lowest concentrations of calcium (*p* < 0.001), phosphorus (*p* < 0.001), and chlorides (*p* < 0.001) and the highest magnesium level (*p* < 0.001) were noted throughout the application of the additives in combination. During the application of sage and EntM separately, only slightly (not significant) increased levels of calcium and magnesium and decreased levels of phosphorus were measured.

No effect of time or treatment was observed on phagocytic activity and oxidative status of the rabbits (Table 6). The values of GPx were not influenced during experimental application of the tested additives (Table 6). All morphometry parameters were influenced by treatment and (except for crypt depth) also by time (Table 6). Throughout the EntM and sage administration, both separately and in combination, the jejunum surface area, VH, CD, and the VH:CD ratio increased to the highest level in group E (*p* < 0.001). All additives, but mostly EntM, showed a tendency to improve the jejunal morhological parameters till the end of the experiment (day 35), with significant increases in villus cut surface (*p* < 0.01), villus circumference (*p* < 0.01), villus height (*p* < 0.01), crypt depth (*p* < 0.05), and VH:CD (*p* < 0.001) compared to control, and VH:CD ratio compared to S (*p* < 0.01), as well.

## 4. Discussion

The antimicrobial effects of bacteriocins and sage extract are well known. Enterocins have a broad antimicrobial spectrum, including both Gram-negative but mostly Gram-positive bacteria; this was also confirmed by us in this experiment. These data correspond to our previously-achieved results during experimental application of several enterocins of rabbit and non-rabbit origin, when strong anti-staphylococcal inhibition effect [4,16,17] and also reduced counts of *E. coli* were detected [4]. Most authors report that Gram-positive bacteria tend to be more sensitive to sage essential oil than Gram-negative ones [41], which could be explained by the stronger membrane of Gram-negative bacteria; these observations are in accordance with our results. Lower counts of coliforms were noted during sage administration alone and in combination with EntM, similarly to Szabóová et al. [19], who reported reduction in *E. coli* due to sage extract administration in rabbits. Coliform bacteria were also inhibited using EntCCM4231 in combination with sage [17] and during administration of lantibiotics nisin and gallidermin to rabbits [13,42]. Kritas et al. [3] also isolated *E. coli* and *Clostridium perfringens* at lower frequency from probiotic-treated rabbits. The decrease in LAB, enterococci, and CoPS noted in the appendix mostly in E + S group at day 21 is accordance with results obtained during lantibiotics application [13,42]; but surprisingly, the coliforms increased in this group in our experiment. To our best knowledge, this is one of the first reports presenting the bacterial composition and/or its changes in the appendix during enterocin application. As information about the microbiota in the rabbit appendix is often rudimentary, monitoring the microbiological changes and alterations during bioactive substance administration can help us to better understand the digestion and immune response of rabbits.

The tested serum parameters were in the range of normal values defined for these parameters in previous studies with rabbits [43,44,45], although there are differences in physiological or reference ranges in rabbit serum. The influence on the lipid and nitrogen metabolism and serum parameters using probiotics, bacteriocins, or herbal extract addition often depends on the period (length) and dosage of their application, the feed composition, and animals’ age and weight [3,4,5,46]. It was interesting to find out that combined administration of EntM and sage extract affected the tested parameters in the opposite way compared to their application separately. Further experiments are needed to confirm these findings, and also to determine in more detail if there is any synergistic and/or antagonistic effect of both additives on the tested parameters. The lowest concentrations of TPs, triglycerids, and glucose and the highest urea and creatinine levels were noted during combined application of EntM and sage extract. While several authors report no influence of nisin, EntCCM4231 and sage addition on biochemical parameters [13,17,19], Ent2019 administration slightly increased the TPs, triglycerids, glucose, and calcium concentrations in our rabbit serum [4]. Bioactive plant compounds may also affect blood parameters by maintaining (positive effect) or altering them (negative effect). The dietary inclusion of EntM elevated the TPs, while sage extract improved the lipid metabolism, reducing HDL and LDL cholesterol, and the LDL/HDL ratio. Antihyperlipidemic or hypocholesterolaemic effects of plants have also been reported [1]. Although the majority of studies mostly present the antioxidant effect of sage, extracts from sage species were also able to reduce plasma cholesterol, LDL, and triglycerides in lipidemic rats [47], similarly to our findings. On the other hand, the significant increase in urea and creatinine levels in the blood during sage addition demonstrates its negative impact on the blood nitrogen profile due to worsening of glomerular filtration in the kidneys. This could be explained by the high astringence effect of sage on the mucous membranes due to its thujone content. After sage extract cessation, the decrease in serum albumin level indicated that the renal function had been normalized. The lower glucose concentrations in the experimental groups could be explained by increased H^+^ concentration due to higher organic acid values in the cecum content, which inhibited gluconeogenesis [48]. This hypothesis was partially confirmed in our case, as only the lactic acid, butyric, valeric, and caproic acids increased during sage application (data not shown). The increased H^+^ (lactate accumulation) in the organism first stimulates physicochemical mineral dissolution by increasing the osteoclast and osteoblast activity (bone resorption) and mostly the Ca^2+^ and Mg^2+^ reabsorption in renal tubules for pH neutralization. At the beginning of the experiment, the calcium concentration in serum was outside the physiological frame, probably due to an unbalanced feed calcium content. During additive application, the calcium level was increased compared to the levels noted at the start of the experiment, and after additive cessation the concentration of calcium decreased again. Redrobe [49] also presented a different form of calcium metabolism in rabbits compared to other animals, because the rabbits´ blood calcium levels fluctuate widely, dependent upon the level of calcium in their diet and the intestinal absorption as well. Another way of explanation could be the activation of parathyroid hormone and release of calcium from bone tissue. Hypermagnesiaemia, hypophosphataemia, and low chloride concentrations in serum at the beginning of the experiment can be caused by unbalanced mineral content in the feed, which was improved during the EntM and sage application. The highest levels of AST and ALT in the E + S group suggest that both bioactive compounds significantly influenced the metabolism in the GIT. This can be explained by increased metabolic processes in the liver due to breakdown of food into energy, e.g., higher TP in serum and increased gluconeogenesis, which is often connected with increased activity of these enzymes, mostly with ALT. Contrary to our findings, decrease in ALT after *Lactobacillus plantarum* probiotic spray application in suckling rabbits was found by Bovera et al. [50]. The beneficial effects of *Salvia officinalis* on lipid profile have also been shown; it could reduce the levels of triglyceride, cholesterol, urea, creatinine, AST, and ALT [51]. Slight decrease in ALT during sage extract application to rabbits was also confirmed by us. We also expected the hypolipidemic influence of nicotinic acid, a component of sage, on the lipase concentrations in serum.

The GPx activity in blood can be a marker of reaction to stress, induced by factors such as diet change and manipulation. EntM and sage inclusion in our rabbits’ diet did not influence GPx values, similarly to Szabóová et al. [17]; we suppose that application of these natural substances did not evoke oxidative stress in the rabbits.

Many studies deal with immunity stimulation, both cellular and humoral, including the stimulation of GALT, mainly by probiotics and plant extracts, but only some of them describe the influence of bacteriocins on non-specific immunity. Although we did not record significant differences in phagocytic activity, the numerical increase indicates a possible immuno-stimulative effect of EntM, in accordance with our previous reports [11,13,52]; moreover, the highest values of PA and IPA were noted in this experiment. Villamil et al. [53] also described the immunomodulatory effect of nisin on non-specific immunity in turbot; their hypotheses are based on receptor competition or excessive accumulation of competent cells. We suppose that our enterocins are able to stimulate the immune system through the modulation of gut microbiota instead of lactic acid bacteria, and by supporting/improving the GALT by stimulating the IgA system. These findings are supported by previous results from several in vivo experiments with enterocins and enterocin-producing and probiotic strains [4,17].

It is known that probiotics are capable of immunity stimulation by improving the intestinal barrier and mucosal immune system through modulation of the intestinal microflora and by production of antimicrobial compounds. Most previous studies deal with dietary treatment effects on intestinal morphology mainly in chickens, pigs, and rats, but data on the rabbit gut morphology are rare [54]. On the other hand, there is new potential for morphometry influence using bacteriocins. Throughout our EntM and sage administration, all the tested morphological parameters in the jejunum were increased compared to control data (except CD, which were decreased), with the highest values in group E. These results are in accordance with those obtained during morphological parameter testing in the rabbit jejunum and ileum using probiotics and prebiotics [55,56], but they are also contrary to nisin application in rabbits, when the jejunal morphology was not influenced [13]. The improved morphometry parameters during additive application suggest better functionality of the intestinal mucosa, more effective intestinal immunity through IgA production, enlargement of the surface area capable of greater absorption of available nutrients, leading to better health status and meat quality, and higher weight gains in rabbits. Of course, development of these results and already-known facts concerning improvement in rabbits’ health, immunity and microbial balance by means of additive supplementation is needed through further investigations.

## 5. Conclusions

In conclusion, it can be stated that some antibacterial effect of EntM was observed against CoNS, and sage extract was effective against CoPS alone and in combination with EntM. The additives reduced the coliforms in the rabbits’ feces. In the appendix, some reduction in enterococci, LAB, and CoPS was observed during administration of both additives (separately and in combination). In the blood, the hypocholesteremic effect of sage alone and in combination with EntM was detected. All additives, but mostly EntM, showed a tendency to improve the jejunal morhological parameters. This study not only has impact for basic research, as one of the first experiments investigating the intestinal morhpological changes in rabbits, but it also helps to clarify and spread knowledge on the interactions between the host organism and natural additives (enterocins and herbal extracts), due to modulation of the gut microflora and immunity.

## Figures and Tables

**Table 1 animals-10-00115-t001:** Ingredients and chemical composition of granulated diet.

Feed Ingredients (%)		Chemical Composition, Minerals and Vitamins (g ^a^, mg ^b^/kg feed)	
Dehydrated lucerne meal	36.0	Dry matter ^a^	882.1
Extracted sunflower meal	5.5	Crude protein ^a^	164.2
Oats	13.0	Crude fiber ^a^	155.5
Wheat bran	9.0	Fat ^a^	33.3
Dry malting sprouts	15.0	Ash ^a^	73.0
Extracted rapeseed meal	5.5	Nitrogen free extract ^a^	466.8
Barley	8.0	Organic compounds ^a^	809.1
DDGS	5.0	Acid detergent fiber (ADF) ^a^	191.1
Sodium chloride	0.3	Neutral detergent fiber (NDF) ^a^	339.5
Premix minerals ^1^	1.7	Lignine ^a^	42.3
Limestone	1.0	Hemicellulose ^a^	148.5
		Cellulose ^a^	148.8
		Starch ^a^	127.2
		Calcium ^a^	6.0
		Phosphorus ^a^	5.9
		Magnesium ^a^	2.3
		Sodium ^a^	1.7
		Potassium ^a^	7.5
		Iron ^b^	403.9
		Zinc ^b^	166.4
		Manganese ^b^	156.7
		Copper ^b^	22.9
		Metabolic energy (MJ/kg)	11.0

DDGS: Dried distillers grains with solubles. ^a^ Ingredients in g/kg feed; ^b^ Ingredients in mg/kg feed; ^1^ Premix contains per kg: Calcium 6.73 g; phosphorous 4.13 g; magnesium 1.90 g; sodium 1.36 g; potassium 11.21 g; iron 0.36 g; zinc 0.13 g; copper 0.03 g; and selenium 0.2 mg. Vitamin mixture provided per kg of diet: Vitamin A 1,500,000 IU; Vitamin D3 125,000 IU; Vitamin E 5000 mg; Vitamin B1 100 mg; Vitamin B2 500 mg; Vitamin B6 200 mg; Vitamin B12 0.01 mg; Vitamin K3 0.5 mg; biotin 10 mg; folic acid 25 mg; nicotinic acid 4000 mg; and choline chloride 100,000 mg.

**Table 2 animals-10-00115-t002:** Bacterial counts (log 10 CFU/g ± SD) in feces of rabbits during enterocin (Ent) M (E), sage extract (S), and their combinative (E + S) application.

Parameter	Day	E	S	E + S	C	SD	Significance of Effects		
							Treatment	Time	Treatment × Time
*Enterococcus* sp.	21	4.86	3.82	4.72	4.02	0.512	0.1130	<0.0001	0.2929
	35	3.07	2.49	2.00	2.00	0.509			
LAB	21	5.04	4.42	4.04	4.39	0.416	0.5198	0.0002	0.1635
	35	3.50	3.05	3.99	3.77	0.405			
CoNS	21	3.26 ^a^	3.57 ^a,b^	3.46 ^a,b^	3.77 ^b^	0.213	0.1094	<0.0001	0.0571
	35	3.75 ^a,b^	3.90 ^a^	3.31 ^a,b^	3.60 ^b^	0.251			
CoPS	21	4.45 ^a^	3.27 ^b^	3.37 ^b^	3.91 ^a^	0.545	0.4838	<0.0001	0.0002
	35	2.18 ^a^	2.73 ^a,b^	3.31 ^b^	2.82 ^a,b^	0.463			
Enterobacteria	21	4.56	4.33	4.60	4.29	0.153	0.1492	0.0003	0.0281
	35	3.91 ^a,b^	3.46 ^a,b^	3.75 ^a^	4.51 ^b^	0.443			
Coliform bacteria	21	2.66 ^a^	1.67 ^b^	2.13 ^a,b^	4.79 ^c^	1.371	<0.0001	<0.0001	<0.0001
	35	1.64	1.56	0.94	1.20	0.326			
*Pseudomonas*-like sp.	21	4.94	4.41	4.55	5.14	0.338	0.0058	<0.0001	<0.0001
	35	3.65 ^a^	5.19 ^b^	3.32 ^a^	3.34 ^a^	0.890			
*Clostridium*-like sp.	21	4.20 ^a^	4.50 ^a,b^	5.12 ^b^	4.54 ^a,b^	0.385	0.5945	0.5911	0.0565
	35	4.64	4.96	4.32	4.84	0.280			

LAB—lactic acid bacteria, CoNS—coagulase-negative staphylococci, CoPS—coagulase-positive staphylococci, sp.—species, ^a,b,c^ Mean values within lines with different superscript letters are significantly different (*p* < 0.05) using by Bonferroni’s post hoc test.

**Table 3 animals-10-00115-t003:** Bacterial counts (log 10 CFU/g ± SD) in appendix of rabbits during E, S, and E + S application.

Parameter	Day	E	S	E + S	C	SD	Significance of Effects		
							Treatment	Time	Treatment × Time
*Enterococcus* sp.	21	4.13 ^a^	1.12 ^b^	2.75 ^c^	5.04 ^d^	1.709	<0.0001	0.2660	<0.0001
	35	4.33 ^a^	3.83 ^a^	2.53 ^b^	3.41 ^a,b^	0.763			
LAB	21	5.10 ^a^	5.10 ^a^	2.97 ^b^	5.10 ^a^	1.605	0.9534	0.0008	0.04994
	35	4.67	4.51	4.24	4.91	0.281			
CoNS	21	3.50	3.41	3.54	3.65	0.099	0.0092	0.0860	0.0185
	35	3.61 ^a^	3.41 ^a^	4.32 ^b^	3.51 ^a^	0.413			
CoPS	21	4.78 ^a^	1.74 ^b^	2.40 ^b^	5.03 ^a^	1.662	<0.0001	0.4953	<0.0001
	35	4.17	3.84	3.09	3.52	0.461			
Enterobacteria	21	5.07	5.07	4.77	5.10	0.156	0.7416	0.0772	0.0581
	35	4.88	4.54	5.11	4.57	0.271			
Coliform bacteria	21	1.91 ^a^	5.25 ^b^	3.33 ^a^	1.83 ^a^	1.602	0.0003	0.1599	0.0086
	35	3.15	3.97	3.38	3.76	0.369			
*Pseudomonas*-like sp.	21	4.45 ^a^	5.02 ^b^	4.20 ^a,b^	4.82 ^a,b^	0.368	0.2664	0.6425	0.0138
	35	5.04	4.50	4.63	4.58	0.241			
*Clostridium*-like sp.	21	5.10	5.07	4.20	4.75	0.418	0.0422	<0.0001	0.8729
	35	7.10	6.85	5.94	6.10	0.565			

LAB—lactic acid bacteria, CoNS—coagulase-negative staphylococci, CoPS—coagulase-positive staphylococci, sp.—species, and ^a,b,c,d^ mean values within lines with different superscript letters are significantly different (*p* < 0.05) using by Bonferroni’s post hoc test.

**Table 4 animals-10-00115-t004:** The effect of E, S, and E + S dietary supplementation on the blood enzymatic and mineral profile of rabbits.

Parameter	Day	E	S	E + S	C	SD	Significance of Effects		
							Treatment	Time	Treatment × Time
Total protein (g/L)	21	51.39 ^a^	49.38 ^a^	44.21 ^b^	48.81 ^a^	3.03	<0.0001	<0.0001	<0.0001
	35	61.49 ^a^	54.81 ^b^	50.40 ^b^	60.63 ^a^	5.21			
Urea (mmol/L)	21	2.79 ^a^	3.78 ^b^	4.84 ^c^	2.72 ^a^	1.00	<0.0001	<0.0001	<0.0001
	35	4.57 ^a^	4.39 ^a^	6.47 ^b^	3.63 ^c^	1.21			
Albumin (g/L)	21	30.65	31.26	31.05	30.45	0.37	<0.0001	0.0103	<0.0001
	35	28.46 ^a^	26.09 ^b^	25.36 ^b^	27.45 ^a^	1.38			
Creatinine (µmol/L)	21	58.56 ^a^	71.29 ^b^	99.57 ^b^	63.25 ^a^	18.37	<0.0001	0.0064	<0.0001
	35	119.60 ^a^	81.25 ^b^	80.20 ^b^	110.40 ^a^	20.15			
Glucose (mmol/L)	21	7.29 ^a^	7.61 ^a^	6.34 ^b^	7.65 ^a^	0.61	<0.0001	<0.0001	<0.0001
	35	6.78 ^a^	6.09 ^b^	4.93 ^c^	6.64 ^a^	0.84			
Triglycerids (mmol/L)	21	1.79 ^a^	2.03 ^a^	1.00 ^b^	2.00 ^a^	0.48	<0.0001	<0.0001	<0.0001
	35	1.44 ^a^	0.95 ^b^	1.09 ^a,b^	1.36 ^a^	0.23			
Cholesterol (mmol/L)	21	2.77	2.42	2.61	2.52	0.15	0.0362	<0.0001	0.0865
	35	1.94 ^a^	1.58 ^a,b^	1.23 ^b^	1.70 ^a^	0.30			
Chol. HDL (mmol/L)	21	0.79 ^a^	0.70 ^b^	0.81 ^c^	0.86 ^a^	0.07	0.0042	<0.0001	<0.0001
	35	0.73 ^a^	0.82 ^a^	0.85 ^a,b^	0.78 ^b^	0.05			
Chol. LDL (mmol/L)	21	1.30 ^a^	0.70 ^b^	0.98 ^c^	1.36 ^a^	0.30	<0.0001	<0.0001	0.0004
	35	0.92 ^a,b^	0.68 ^a^	0.84 ^a,b^	1.02 ^b^	0.14			
LDL/HDL	21	1.63 ^a^	1.01 ^b^	1.24 ^b,c^	1.57 ^a^	0.29	<0.0001	<0.0001	<0.0001
	35	1.27 ^a^	0.84 ^b^	0.99 ^c^	1.33 ^a^	0.2323			

HDL—high-density lipoprotein, LDL—low-density lipoprotein, and ^a,b,c^ mean values within lines with different superscript letters are significantly different (*p* < 0.05) using by Bonferroni’s post hoc test.

**Table 5 animals-10-00115-t005:** The effect of E, S, and E + S dietary supplementation on the blood protein and energetic profile of rabbits.

Parameter	Day	E	S	E + S	C	SD	Significance of Effects		
							Treatment	Time	Treatment × Time
AST (µkat/L)	21	0.64 ^a^	0.99 ^b^	1.52 ^c^	0.55 ^a^	0.44	<0.0001	<0.0001	<0.0001
	35	1.86 ^a^	0.79 ^b^	0.43 ^c^	0.98 ^d^	0.61			
ALT (µkat/L)	21	0.86 ^a^	0.66 ^b^	1.44 ^c^	0.77 ^b,c^	1.71	<0.0001	<0.0001	<0.0001
	35	2.66	2.08	1.76	2.03	0.38			
GMT (µkat/L)	21	0.19	0.16	0.21	0.18	0.02	0.4322	0.0001	0.8396
	35	0.08	0.09	0.14	0.13	0.03			
ALP (µkat/L)	21	1.85 ^a^	1.78 ^a^	1.43 ^b^	2.15 ^c^	0.30	<0.0001	<0.0001	<0.0001
	35	1.18 ^a^	1.36 ^a,b^	0.84 ^c^	1.47 ^b^	0.28			
Total bilirubin (µmol/L)	21	6.44 ^a^	5.91 ^a^	3.13 ^b^	3.78 ^b^	1.61	<0.0001	<0.0001	<0.0001
	35	3.94	3.66	4.26	3.30	0.41			
Lipase (µkat/L)	21	1.47	1.31	1.53	2.01	0.30	0.9989	0.9669	1.0000
	35	1.34	1.42	1.52	1.37	0.08			
Alfa-amylase (µkat/L)	21	5.13 ^a^	6.06 ^b^	4.70 ^c^	5.51 ^a^	0.58	<0.0001	<0.0001	<0.0001
	35	4.51 ^a^	5.59 ^b^	3.50 ^c^	4.58 ^a^	0.85			
Pancreatic amylase (µkat/L)	21	4.43 ^a^	5.20 ^b^	3.27 ^c^	4.34 ^a^	0.79	<0.0001	<0.0001	<0.0001
	35	4.46 ^a^	4.96 ^b^	4.51 ^a^	4.32 ^a^	0.28			
Ca (mmol/L)	21	2.32 ^a^	2.31 ^a^	1.43 ^b^	2.24 ^a^	0.43	<0.0001	<0.0001	<0.0001
	35	1.76 ^a^	1.39 ^b^	1.53 ^c^	1.55 ^a^	0.15			
P (mmol/L)	21	2.43 ^a^	2.19 ^b^	1.91 ^c^	2.45 ^a^	0.25	<0.0001	<0.0001	<0.0001
	35	1.92 ^a^	1.56 ^b^	1.88 ^a^	2.49 ^c^	0.3923			
Mg (mmol/L)	21	0.74 ^a^	0.78 ^a^	1.19 ^b^	0.67 ^a^	0.23	0.0353	<0.0001	0.0107
	35	0.46	0.65	0.57	0.61	0.0823			
Chlorids (mmol/L)	21	76.87 ^a^	80.63 ^a^	66.00 ^b^	79.63 ^a^	6.71	<0.0001	<0.0001	<0.0001
	35	72.25 ^a^	61.50 ^b^	61.13 ^b^	72.00	7.2223			

AST—aspartate aminotransferase, ALT—alanine aminotransferase, GMT—gamma glutamyl transferase, ALP—alkaline phosphatase, and ^a,b,c,d^ mean values within lines with different superscript letters are significantly different (*p* < 0.05) using by Bonferroni’s post hoc test.

**Table 6 animals-10-00115-t006:** The effect of E, S, and E + S application on phagocytic activity of leukocytes, glutathione-peroxidase activity, and jejumal morphometry of rabbits.

Parameter	Day	E	S	E + S	C	SD	Significance of Effects		
							Treatment	Time	Treatment × Time
Phagocytic activity	21	64.15	61.52	62.50	61.17	1.35	0.6986	0.6772	0.8203
	35	63.00	63.83	63.00	61.67	0.89			
Index of phagocytic activity	21	2.04	2.33	2.27	2.00	0.17	0.9384	0.3196	0.4799
	35	2.52	2.25	2.18	2.32	0.15			
Glutathione-peroxidase (µkat/L)	21	191.2	201.8	198.0	168.5	14.94	0.2390	0.0077	0.6971
	35	205.5	252.3	225.3	217.4	19.89			
Villus cut surface (µm^2^)	21	84,600.0 ^a^	83,222.0 ^b^	83,805.0 ^a^	81,582.0 ^c^	1278.7	<0.0001	0.0264	0.2902
	35	83,592.0 ^a^	82,425.0 ^a,b^	82,877.0 ^a,b^	81,871.0 ^b^	728.0			
Villus circumference (µm)	21	1627.0 ^a^	1602.0 ^b^	1611.0 ^a,b^	1572.0 ^c^	23.9	<0.0001	0.0214	0.3833
	35	1608.0 ^a^	1587.0 ^a,b^	1596.0 ^a,b^	1575.0 ^a,b^	13.9			
Villus height (µm)	21	711.0 ^a^	699.0 ^b^	704.0 ^a,b^	686.0 ^c^	10.6	<0.0001	0.0201	0.2853
	35	702.0 ^a^	693.0 ^a,b^	696.0 ^a,b^	688.0 ^b^	5.9			
Crypt depth (µm)	21	175.0	180.0	178.0	183.0	3.0	0.034	0.2563	0.9606
	35	177.0 ^a^	182.0 ^a,b^	179.0 ^a,b^	186.0 ^b^	3.9			
Villus height:crypt depth	21	4.03 ^a^	3.87 ^b^	3.97 ^a,b^	3.75 ^c^	0.13	<0.0001	0.0127	0.9895
	35	3.98 ^a^	3.82 ^b^	3.90 ^a,b^	3.70 ^c^	0.12			

^a,b,c^ Mean values within lines with different superscript letters are significantly different (*p* < 0.05) using by Bonferroni’s post hoc test.

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
