# Peer review of "Can Enterocin M in Combination with Sage Extract Have Beneficial Effect on Microbiota, Blood Biochemistry, Phagocytic Activity and Jejunal Morphometry in Broiler Rabbits?"

_animals, 2020, doi:10.3390/ani10010115_

Round 1

Reviewer 1 Report

Despite the authors addressed some concerns, but some point still needs to be fixed to be fit for publication:

Table 2: it is impossible to analyze data of only one sample. In this case, this data at (0 day) should be removed. Line 166: Gancarčíková et al. [26] is not the proper reference for the methods used. The authors must add the original reference for each parameter. For example alanine aminotransferase (ALT), aspartate aminotransferase (AST) were determined following the methods of Reitman and Frankel (1957) and Kind and King (1954), respectively. Tables 4 and 5 the full term of the abbreviations used must be clarified in the footnotes of the tables like ALT, AST, GMT. The titles of all tables are unclear. I think " The effect of enterocin M (E) and/or sage extract (S) dietary supplementation on................of rabbits" would be more appropriate.

Author Response

Response to Reviewer 1 Comments

Reviewer 2 Report

December, 30

The last version of the manuscript "Can have enterocin M in combination with sage beneficial effect on microbiota, blood biochemistry, phagocytic activity and jejunal morphometry in broiler rabbits?” has been significantly improved. However, there are still several inaccuracies in the text, that authors must correct.

Abstract

Page    line

1          29-31

“Dietary supplementation of EntM and sage significantly reduced the coliforms (E: p < 0.001; S: p < 0.001; E+S: p < 0.001) in faeces. Reduced bacterial counts, except for coliforms in the appendix, were noted in the E+S group”

Change to

“Dietary supplementation of EntM and sage significantly reduced the coliforms (E: p < 0.001; S: p < 0.001; E+S: p < 0.001) in faeces, while simultaneous addition of EntM and sage decreased enterococci (E+S: p < 0.0001), lactic acid bacteria (E+S: p < 0.01), and positive staphylococci (E+S: p < 0.0001) in the appendix”

Page    line

1          33-36  

NOTE, in table 6, that,

For TP: 51.38 and 49.38 have the same letter (a) to 48.8.

For PA: 64.17, 61.50 and 62.5 were not different of 61.17 (p=0.69)

For IPA: 2.03, 2.33 and 2.27 were not different of 2.0 (p=0.93)

Therefore, authors cannot say that EntM, sage or combination increase the TP and PA activity.

And for that, the following sentence is not true: 

 “ EntM application increased the total protein concentration and phagocytic activity (PA), and the morphological parameters were improved (p < 0.001). Sage alone and in combination with EntM also stimulated PA and jejunal morphometry (p < 0.01) in rabbit”

Please, the sentence must write to:  

“EntM and sage application, alone or in combination, improve the jejunal morphometry (p < 0.0001) in rabbit”

Keywords

OK

Introduction

OK

Materials and methods

The data came from males (n=60) and females (n=4), therefore the authors have to two possibilities:

i) to include the sex as fixed effect in the model,

or ii) to eliminate all records of the females.

Results

Page          line

5                200-201    Insert “in faeces” …In faeces, the bacterial counts (except Clostridium-like sp.) were influenced by time.

5                204                       Change “day 21” at the end of sentence “  …… and coliforms (E: p < 0.001; S: p < 0.001; E+S: p < 0.001) was noted in faecal samples compared to C during EntM and sage application on day 21.

5                206                       Delete “The combination of both additives affected CoPS and coliforms in faeces” because this information is indicated before.

5                209                       Insert the S groups for CoPS “…and CoPS (S: p < 0.001; E+S: p < 0.001),….”

5                212-215    Among blood biochemical parameters (Tables 4 and 5), indicators of lipid metabolism (triglycerids, cholesterol, HDL and LDL/cholesterol) already exceeded the physiological levels reported for these parameters at the beginning of the experiment; however, the chloride level was reduced compared to the control data.

This sentence is not support by data of table 4 and 5, when we see the p_values among E, S, E+S and control group, the values of S and E+S were lower than control; or authors have others initial values on day 0?. Therefore, authors must delete this sentence, or they give the initial values.

5                216                       Delete “where there was only time effect) “

5                216                       Inserte “…..and lipase (Table 4 and 5)

5                217-219                Rewrite “The most striking differences in lipid metabolism were observed during the administration of sage extract alone, but also in combination with EntM; when in this regard, sage extract alone or together with EntM reduced the level of HDL cholesterol (E: p  < 0.05; S: p < 0.001; E+S: p <0.05), LDL cholesterol (S: p < 0.001; E+S: p < 0.001) and LDL/HDL ratio (S: p < 0.001; E+S: p < 219 0.01) , compared to in comparison with control data”.

5                220-221                Moreover, the trend towards decrease in lower LDL cholesterol level and LDL/HDL ratio was were still evident in groups S (p < 0.01; p < 0.001) and E+S (p >  < 0.05; p < 0.001) even two weeks after additives cessation (day 35).

5                222-224                Delete “Even though the application of EntM alone only slightly affected the lipid metabolism, on the other hand numerical increase in TP concentrations was noted, which was the highest of all tested groups.”, because the values in TP, triglycerides, cholesterol, HDL, LDL and LDL/HDL in EntM were not significative difference from control group.

5                231-233                Delete “ALT and total bilirubin” because there are no differences between EntM and control groups, or the comparation is between EntM and E and S? in the last case, authors must indicate.

6                278                       Rewrite the sentence “…..judging from the higher TPs in group E and higher TPs, albumin and triglycerids in group S,…..” because this affirmation is no true. Table 4 shows similar valuer with control group.

7                318                       The superscript letters indicate similar values of S and E+S to Control, then S did not decrease ALT level, but E show higher level to S, E+S and control.

Conclusions

Page          line

8                355                       Change “…..enterobacteria in combination….  ” to “    …CoPS in combination    “

8                357-360    The sentence must be deleted because it is based in a wrong interpretation of results.

Note, Control group is the reference group. Then, table 4 shows similar TP concentration among E, S and C groups, and only TP concentration is lower in E+S vs control. For ALT, S and E+S show similar values to control, only E group shows higher levels than control group. The superscript letters are right???.

Table 2 and 3              Change “CoNS- coagulase-positive staphylococci” to “CoNS- coagulase-negative staphylococci”

Author Response

Response to Reviewer 2 Comments

This manuscript is a resubmission of an earlier submission. The following is a list of the peer review reports and author responses from that submission.

Round 1

Reviewer 1 Report

The new version of the manuscript intitled "Can have enterocin M in combination with sage beneficial effect on microbiota, blood biochemistry, phagocytic activity and jejunal morphometry in broiler rabbits?” has been significantly improved. However, I propose the following recommendations for authors.

Page  line
4 177-178 Please, change “The effect of treatment was observed only in the clostridia and pseudomonads; however, a treatment x time interaction was also found (Table 2)” to “The effect of treatment was observed only in the coliform and pseudomonads; treatment x time interaction was also found significant in these groups of bacteria (Table 2)”. Note that p=0.59 for clostridia (NS).

4 179 For CoPS, table 2 shows that the means 3.27 and 3.37 have the letter “b” and the mean 3.91 has the letters “ab”; thus 3.27 and 3.37 were not different from 3.91.
Therefore, authors must delete “CopS (S: p < 0.01; E+S: p < 0.01)”

4 180 See the last comment. Then, authors must delete “pseudomonads (S: p<0.05) and clostridia (E: p < 0.05)” (I see the table and there are no differences)

4 181-182 For CoPS, 2.18 has the same letter that 2.82 and, for enterobacteria, 3.46 has letter ”a” as the value 4.51. Then, the sentence has to be rewritten to “ ……..lower counts of enterobacteria (E+S: p < 0.05) were detected than in group”

4 186 There is also difference in enterococci between E vs C groups.
See before comment, the group C shows a value of 4.67, and it has letters “ab” then there are no significant differences in enterobacteria, among E, S, E+S to C groups. Therefore, author must delete “enterobacteria (E: p < 0.05; S: p < 0.05)”

Please, confirm in table 3 that in Enterococcus, 2.75 and 5.04 have letter “c”. If they have the same letter, they are no difference between them.

5 192 Please, insert “and” (Table 5 and 6).

5 208 Please, rewrite “….while during its application the levels of albumin in blood remained high “. It is incorrect to say “increased” because of they have same letter.

5 211 Please, delete GMT because values are no different between treatments.

5 216-217 Please, confirm the affirmation “…………..During the application of sage and EntM itselves, increased levels of calcium and magnesium and decreased levels of phosphorus were measured…… “ because this affirmation is no correct when I see the letters of the means.

5 219-224 If there are no effect of treatment (p = 0.69 and p=0.92 in PA and oxidative status) and time (p=0.67 and p=0.31), the authors cannot argue that there are differences between treatments and time. Then, “At day 21, the EntM showed the most stimulative effect on unspecific imunity parameter, when the highest PA was measured in E group, followed by the E+S group. While sage application itself only slightly increased the PA during its inclusion, 2 weeks after the additives cessation (day 35, the end of the experiment), the highest PA value was detected just in group S. PA values in all experimental groups were almost at the same level at day 35 and they were still higher than in C.” has to be deleted.

Discussion and Conclusion Sections Please, authors must review the discussion and conclusion sections, after they rewrite the Results Section

For example, there are no differences between treatments and control groups for PA and IPA; then, the authors cannot argue that “The trend for increase in the PA and IPA values during bacteriocins application and the prolonged immuno-stimulative effect were observed still also after their cessation, in accordance with our previous reports [11,13,37]”

Tables Note that there are 2 tables number 6.
In tables 5 and 6, authors must change “,” for “.”

Author Response

 Response to reviewer 1 Comments

Reviewer 2 Report

The searching for safe natural feed additives in the rabbit industry has paid growing interest in recent years. Hence, the topic of manuscript is of interest but several concerns need to be addressed to be fit for publication as follows: Critical comments: 1. A major concern in this study is the addition of these feed additives in the water. Firstly, the authors must measure the water consumption per rabbit to determine the actual amount of feed additives reach each rabbit. How the authors be sure that the proper amount of feed additives has to be given to each rabbit? What about the solubility of these additives in the water? What is the drinking method used nipple or potteries? I think if the authors related the concentration used to the water amount (i.e. the concentration of additives/ liter) would be more applicable. 2. The statistical analysis is not appropriate to the design of the study as the authors used two sexes but have not incorporated the sex as a factor in the statistical analysis of the findings. 3. Freshly-voided faeces were collected using nets mounted under the cages. How the authors have assured the source of microbes is mainly related to the gut not from the surrounding environment? What is meant by freshly? 4. "Results concerning caecal bacteria have been already presented in the journal Probiotics and Antimicrobial Proteins by Pogány Simonová et al., 2019, https://doi.org/10.1007/s12602-019-367 09584-z". Thus, why this data "table 3" is republished here??? I think this is unacceptable. The same situation for the growth performance as it should not be mentioned in the results section. 5. Tables 5 and 6: SD for 0 days for all parameters is 0. This means one replicate has been used, thus this could not be statistically analyzed. Specific comments: 1. The title: change sage into the sage extract. 2. Simple summary and conclusion: add a conclusion of the main findings of the study. 3. Introduction: - Line 42-55: the authors describe the effects of probiotics in rabbit feeding which is not related to the study. Please remove these paragraphs. - More information on the sage extract should be added. - Line 66-67: 8 references is a very large number. Please, refer to the most suitable references. - The hypothesis of using these additives should be clarified before the aim of the study. 4. Material and methods: - Line 74: please add the mean initial body weight. Also, both sexes should be detailed of the exact number of males and females used. - Line 75: three groups but in tables 4 groups. Please revise. - "A cycle 16 h of light, 8 h of dark" indicates that this experiment was carried out in summer. Thus a temperature of 16 ± 4ºC" is somewhat strange. Please, revise and mention when this study was conducted. - From day 0 to day 21 but in tables 35 days. Please, revise. - All analytical studies in blood should be cited by proper references. 5. Results: - Lines 225-231: the figures of the histological sections should be added. - Table 1: have the authors formulated the diet or used a commercially prepared one??. If they used a commercial one, what is the source of this data? Also, the chemical analysis is calculated or analyzed. If, it is calculated, on what basis??? If it is analyzed, mentioned the methods in the material and methods section? 6. Discussion: The authors must clearly describe the outcome of the combination of both additives.

Author Response

Response to Reviewer 2 Comments

Round 2

Reviewer 2 Report

The authors haven't positively responded to most comments and nearly all required modifications have not been performed. The manuscript still needs extensive revisions and all modifications should be corrected in the manuscript not only in the repose letter. The authors need to revise the earlier comments again point by point with a special focusing on the following points:

Provide a reference confirming the solubility of these compounds in water. For the practical application of the findings of the study, the authors should relate the concentration used to the water amount (i.e. the concentration of additives/ liter). The problem of the statistical analysis must be solved by incorporating the sex factor as a random effect in the statistical model used. Ethically, the results of the caecal bacteria and growth performance must be removed from the material and methods and results section as it is previously published. Tables 5 and 6: it is impossible to analyze data of only one sample. In this case, this data at (0 day) should be removed. Both sexes should be detailed of the exact number of males and females used. Even if the authors used commercial kits for blood biochemical analysis, the references for the methods must be added. The author's response regarding the figures of the histological sections is not convincing. This data must be added to proof of the obtained results. The manufacture of the commercial diet never mentioned the exact formulation thus how the authors exactly know the diet formulation in table 1???? Also, in scientific research, we could not depend on the analyzed composition of the manufacturer. Thus, the authors either analyze the diet or calculate the chemical composition with appropriate references for all ingredients not only for metabolizable energy only.